# Killing the Comments: Why Do News Organizations Remove User Commentary Functions?

**Maria N. Nelson** [1,*], **Thomas B. Ksiazek** [2] and **Nina Springer** [3]

1 Mixed Methods Research Lab, Department of Family Medicine and Community Health, Perelman School of Medicine, University of Pennsylvania, Philadelphia, PA 19104, USA
2 Department of Communication, Villanova University, Villanova, PA 19085, USA; thomas.ksiazek@villanova.edu
3 Department of Communication, University of Münster, 48143 Münster, Germany; nina.springer@uni-muenster.de
* Correspondence: maria.nelson@pennmedicine.upenn.edu

**Abstract:** User commentary in digital journalism is commonly understood as a form of public user engagement and participation, a stance that reframes news organizations' role as discussion curators as necessarily consequential. Yet, in recent years many news organizations have limited, or abandoned altogether, their commentary functions. This paper examines statements and policies published by such news organizations. Based on a thematic analysis of 20 comment removal statements, we found that the most common rationale for this shift was an effort to reduce incivility and misinformation among user comments. The statements analyzed also indicate that organizations are moving to outsource commentary to social media platforms. Tapping into normative discourses of (avoiding) uncivil, conspiracy-prone commentary seems to be an acceptable rationale for abandoning infrastructures established for public discussions or to move these to social media; yet, we found no reflection whatsoever about the additional power afforded to social media companies through such a shift.

**Keywords:** journalism; user comments; incivility; public discourse; online news; comment feature removal



## 1. Introduction

Digital journalism 2.0 used to provide its readers the opportunity to engage with and debate the news of the day, especially stories related to politics, using the same virtual platform with which the information is accessed (Boczkowski and Mitchelstein 2012; Ruiz et al. 2011). The ease of acquiring and engaging with news stories was at first largely viewed as conducive to the type of deliberation required for democratic public engagement, but concerns about the quality of such comments emerged soon thereafter (e.g., Goodman 2013; Ksiazek 2018; Ksiazek and Springer 2020; Liu and McLeod 2021). Serious anxieties in the news industry about the quality of user comments had been well documented over time. For instance, Wolfgang (2018) found that, "journalists disliked moderating conversations and spoke about [reader] contributions derisively, therefore establishing low expectations of the commenter . . . [and] describing the forums using terms like 'cesspool' or 'fetid swamp'" (p. 27). Meltzer's (2015) analysis of more general industry discourse about user comments also found widespread concern among news professionals about incivility in comments. Similarly, in Goodman's (2013) study for The World Association of Newspapers of 100+ news organizations across 63 countries, journalists expressed concerns about the general quality of the online discussions, as well as the resources necessary to curate more productive discussion spaces.

All of this suggests a desire among news professionals to avoid further contributing to widespread social and political polarization across the globe. While some research suggests that commenters are "inclined to seek politically dissimilar conversational partners"

(e.g., Liang 2014, p. 487), a pattern that contributes to viewpoint diversity in comment spaces, these platforms also have the potential to enable group polarization, especially if commenters act in uncivil ways toward one another.

In order to tackle the problem of uncivil user commentary, many news organizations started early on to regulate comments according to specific criteria (Coe et al. 2014; Diakopoulos and Naaman 2011; Ksiazek 2015; Meltzer 2015; Post and Kepplinger 2019; Wolfgang 2018); in a number of instances, this concern has now finally resulted in the removal or severe curtailing of commenting functions on news websites (Goodman 2013; Liu and McLeod 2021). In fact, Liu and McLeod (2021) note, "Many major news outlets online have removed their on-site comment systems. The websites involved concerned different types of media, including wire services (e.g., Reuters), radio stations (e.g., NPR), cable news channels (e.g., CNN), newspapers and news magazines (e.g., Chicago Sun-Times, The Boston Globe, and The Week), and science/technology news outlets (e.g., The Verge and Popular Science)" (p. 868).

Another common strategy is the outsourcing of user discussions to the outlets' social media presences (e.g., Stroud et al. 2020). Thus, in the most recent years, we seem to be returning to a state of digital journalism resembling the less interactive era of "Web 1.0", rather than tackling the perceived problem head-on by exploring ways to encourage more thoughtful contributions and discussions.

Research suggests that commenters and comment readers oppose the trend toward shutting down comment spaces (e.g., Liu and McLeod 2021; Stroud et al. 2020). Given these expectations, the decision to strip away an outlet-loyal community from the ability to participate in the interpretation and discussion of news via commenting on their preferred news website (e.g., Singer et al. 2011) calls for an understanding of the organizational logic behind this shift. This is particularly consequential since most news outlets have a "dual nature" (Plantin et al. 2018, p. 297): They represent a democratically relevant institution, journalism, yet depend on commercial considerations and therefore have to manage participation in a profitable way. Similarly, the social media platform to which commenting is most often outsourced—Facebook—binds "pre-defined communicative acts to an economic logic" (p. 297). News media ought to understand, critically reflect and actually report on the fact that social media's goal "of gathering users' personal data determines the technical properties of platforms, which in turn shapes how they organize communication among users" (p. 297), and that monopolizing, i.e., "achieving lock-in is among platform builders' principal goals" (p. 298). Accordingly, this paper seeks to explore how restricting, shutting down, or moving this participatory feature is communicated to news outlets' users.

### 1.1. News Outlets as Governors and Gatekeepers of User Discussions

From the perspective of infrastructure studies and platform studies, commercially oriented news outlets have a similar "dual nature" as commercial digital platforms whose "affordances support innovation and creativity—supplying a base for video games or new media forms—yet simultaneously constrain participation and channel it into modes that profit the platform's creators" (Plantin et al. 2018, p. 297). However, in digital journalism, another aspect must be considered as contributing to a multifaceted nature: Since the inception of digital journalism, news organizations have wrestled with the tension between encouraging new forms of user engagement, while keeping costs at bay, and maintaining their traditional journalistic authority as information gatekeepers. In the field of journalism studies, scholars have aimed to capture the phenomenon of greater user involvement in the journalistic process through frameworks such as "participatory journalism" (e.g., Singer et al. 2011), "citizen journalism" (e.g., Chung et al. 2018), or "reciprocal journalism" (e.g., Lewis et al. 2014). Whether or not digital journalism has ever truly embraced users as part of the process is a topic of extensive debate. Independent of how one answers this question, news outlets started to create infrastructures for public discussion of the news, along with mechanisms for governing how this infrastructure is used.

That news organizations set community guidelines and expectations, implement and enforce policies to manage users and their discussions, and moderate comment spaces are all illustrative of a strong reluctance to cede control of authority as purveyors of news and information. On most, if not all, news websites that allow commentary, users are expected to abide by published community guidelines. This comment management approach signals a more hands-off strategy, where users are often simply trusted to comply and action is taken only when abuses are reported. A more active approach involves required and/or enforceable policies to manage commenters. These include requiring user registration (whereby the organization maintains a profile record with personal details about the user), anonymity policies (i.e., some outlets allow anonymous commentary, while others prohibit it), reputation management systems (where users can rate, rank, like, etc. other users) and pre-/post-moderation protocols (e.g., profanity filters and retroactive removal of abusive comments) (Braun 2015; Coe et al. 2014; Diakopoulos and Naaman 2011; Díaz Noci et al. 2010; Ksiazek 2015, 2018; Ksiazek and Springer 2020; Ruiz et al. 2011; Santana 2014, 2019).

News organizations widely employ comment moderators, both staff and algorithmic. Stroud et al. (2016) found that 61% of US news organizations employ staff specifically to moderate user comments. However, human moderation can be resource intensive, and in some cases this work is done by journalists, not dedicated moderators (Goodman 2013). Many news organizations share the moderation burden between human moderators and algorithms. While basic profanity filters have been common for some time as a tool used alongside active human moderation, the latest endeavors are experimenting with Artificial Intelligence, and more specifically with machine learning, to automatically detect and filter out disruptive or poor-quality comments (e.g., Schabus et al. 2017; Stoll et al. 2019).

When news organizations moderate comments, they enact a longstanding power structure that privileges the organization and its journalists over users. In the aforementioned survey by the World Association of Newspapers (Goodman 2013) of 104 news organizations from across the globe, there was "general consensus that . . . it is up to the publication to determine the kind of conversation it wants to host" (p. 7). In a particularly illustrative quote, *Die Zeit* (Germany) suggested, "it's absolutely up to you as a newsroom to control what sort of comments you want to have" (p. 8). In Wolfgang's (2018) ethnography of comment moderation inside a US news organization, he observed that newsroom staff "moderated silently and without sincere interaction with commenters in order to maintain journalistic authority" (p. 27). The desire for control over comment spaces suggests that news organizations across the globe view themselves as gatekeepers of user commentary.

## 1.2. Styles of Governing User Commentary

How exactly to govern or manage user comments entails consecutive decisions: (1) should comments be *displayed* or *abandoned* on the outlet's website, or (2) should they be *outsourced* to the outlet's social media presence (and thereby decoupled from the original news production process)? If comments are made visible on the website or social media presence, comments can be moderated in an *engaging* or *policing* fashion (Ksiazek and Springer 2020; Ziegele and Jost 2020; Ziegele et al. 2018). Engaging moderation is the most constructive, but also resource-intensive style of comment management: An *engaging* approach is characterized by the active presence of moderators and/or journalists in comment threads. Moderator engagement can be supportive/rewarding, but of course also regulative/sanctioning (Ziegele and Jost 2020; Ziegele et al. 2018). Supportive moderators can introduce a topic, provide (additional) information, answer questions, or ask users to elaborate interesting claims. Taking on a more general role of "community managers", moderators can also mediate between editors and users. In a regulative role, they can ask users to remain on the topic, be respectful, or mediate conflicts among users.

Research on engaging moderation styles suggests that this form of engagement can positively impact readers' willingness to participate (Ziegele and Jost 2020), decrease incivility, and result in a greater provision of evidence in comments (Stroud et al. 2015; see also Ziegele et al. 2018). However, news outlets are advised to engage in an appropri-

ate way: Regulative (patronizing) moderation can lead to more uncivil user reactions (Ziegele et al. 2018). In addition, experimental research shows that while a sarcastic moderation style might be more entertaining to read, it can negatively affect credibility assessments, thereby quality perceptions, and thus reduce the willingness to participate (Ziegele and Jost 2020). At the same time, these authors also found that factual moderation of uncivil user comments conveys a deliberative discussion atmosphere, which is appealing to users and thus stimulates the participants' willingness to engage. The latest research efforts focusing on automation seek to identify constructive commentary in a more resource-efficient manner (e.g., Haim et al. 2019; Häring et al. 2018; Park et al. 2016; Schabus et al. 2017). Once such comments are identified, they can be featured prominently (e.g., the New York Times "picks"), and this type of recognition might positively impact users' future commenting behavior. In addition, comments can contain useful information for journalists or moderators to consider; an automated identification of such elements (e.g., user feedback or valuable leads for follow-up stories) makes it less resource-intensive for newsrooms to process and include this information in their daily workflows (e.g., Park et al. 2016, p. 1117; see also Häring et al. 2018).

A more restrictive style of comment management, *policing*, is often more cost effective, and therefore more common, than an engaging style (e.g., Haim et al. 2019; Ksiazek and Springer 2020; Ziegele and Jost 2020; Ziegele et al. 2018). A policing approach to comment moderation and management involves identifying and deleting abusive comments and/or content that violates community guidelines. Here, moderation remains reactive or hidden from users. In the worst case, the post is removed (post-moderation) or never published (pre-moderation), and the user does not get informed about this intervention and never learns why their comment was deemed unacceptable. In the best case, moderators inform a commenter about reasons for why a comment had not been published; and, in order to also inform others, such banning can be communicated publicly (e.g., instead of publishing the banned comment, a message can be posted that "this comment violated our community guidelines").

In contrast to engaging and policing comment management and moderation approaches, both of which signal varying investments of organizational resources to manage user discussion spaces, a growing number of news organizations are choosing to *abandon* user commentary functions, altogether, or *outsource* them to social media platforms. There is a clear need to better understand this recent trend toward disabling user commentary features, a form of user engagement that has consistently been championed as an opportunity for user empowerment and participation in digital journalism. This study aims to address this need and is guided by the following research question: *What are the reasons and rationales provided by news organizations for removing or limiting user commentary functions on their digital platforms, or outsourcing them to social media?*

## 2. Methods

In order to better understand the decisions of different news organizations to remove, limit, or outsource commentary functions, we collected and analyzed statements and policies announcing or describing these changes. The first stage in the analysis process involved identifying which news organizations recently made the decision to remove comment functions housed on their online platforms. This purposive sample was collected in April and May of 2019 by a combination of the following procedures. First, we combed through lists of the most popular news organizations compiled by the Nieman Journalism Lab and the Pew Research Center. Then, we searched the following major trade publications and associations for articles about news organizations removing their commentary features: American Journalism Review, Association for Education in Journalism and Mass Communication, Broadcasting and Cable, Columbia Journalism Review, Editor and Publisher, Journalist's Resource, MediaPost, Media Week, Multichannel News, National Association of Broadcasters, Radio Ink, Society of Professional Journalists, and TV NewsCheck. Finally, we conducted a LexisNexis search for articles about commentary

removal using the search terms remov* comment* news[1] to search all lists, organizations, publications, and databases to identify articles mentioning comment removal. After identifying news outlets that had moved or removed commenting functions from their digital platforms, published statements regarding this decision were collected.

Though many digital news outlets released statements describing the decision to remove commenting capabilities from their online platforms, many others made no effort to inform their readers as to why commenting was no longer available. Out of the 28 national and local news outlets identified as having removed their commenting functions from their websites, only 15 were found to have published statements or articles regarding the change, with an additional five included based on the grounds that their commenting policy referenced the removal or limiting of commenting functions. During analysis, depending on the query involved, these five outliers were included or removed from the dataset in order to gain a more holistic sense of the framing of news organizations' decision to end, or severely curtail, commenting functions for their online articles (see Table 1 for a list of the news organizations included in the comment removal analysis).

**Table 1.** News organizations included in comment removal analysis.

| | | |
|---|---|---|
| • Above the Law | • NPR | • The Guardian |
| • BBC | • Popular Science | • The Mic |
| • CNN | • Recode | • The Verge |
| • Hartford Courant | • Reuters | • The Week |
| • Huffington Post | • The Atlantic | • USA Today |
| • New York Times | • The Daily Dot | • Vice |
| • Newsday | • The Daily Beast | |

Outlets that removed commenting feature, but had no published statement available for inclusion in the analysis: Atlanta Journal Constitution; Bloomberg; Buffalo News; Chicago Sun Times; Chicago Tribune; MSNBC; New York Daily News; The Washington Inquirer

The scope of this analysis includes news organizations headquartered in the United States and the United Kingdom. The analysis draws on both automated text analysis (text mining; keyword frequency analysis) and manual qualitative analysis using modified grounded theory (Strauss and Corbin 1990) to identify themes in the organizational reasoning for removing comments. The text analysis functions found in Nvivo 12 that tally all incidences of any given word or phrases allow for a clear sense of the language news organizations use to describe their comment removal process in their statements. Reflecting a growing trend in the discipline, this research design combines both manual and automated text analysis.

Upon identifying the 20 relevant statements and policies, the text of each was uploaded to the qualitative analysis software Nvivo 12. A codebook was iteratively developed using a modified grounded theory approach in order to identify common themes and language (see Supplementary Materials). The codebook was continually refined and revised throughout the analysis phase in order to reflect the researchers' evolving understanding of the topic and to fully reflect the language, themes, and common characteristics of the statements and policies in question. Key parent nodes included references to good commenting practices, bad commenting practices, and moderation techniques or explanations. These nodes were cross-coded to capture specific language used to describe comment function removal, civil and uncivil discussion, and news organizations' ideal community of online readership. Some of these associations were veiled or implicit, while other writers condemned specific behaviors, ideologies, or motivations. Due to the use of a combination of both automated text analysis and manual qualitative analysis, the use of a single coder was sufficient to identify salient themes in a collaborative effort involving the authors.

Considering the timing of statements in relation to a wider trend toward the dissolution of commenting functions housed on news sites, the statements ranged in date, with publication years falling between 2013 and 2018. Five statements were released in 2015,

while four were released in 2016, and four more had no date associated with them. Though this information does at first seem to reveal an interesting trend revolving around the run-up to and aftermath of the 2016 U.S. presidential election, it is important to note that some websites catalogue the date that the statement was last updated or edited, rather than displaying the static initial publication date.

## 3. Results

Published statements on comment removal tended to follow a similar arc. Many began with a description of the toll that comment section moderation and poor-quality commenting practices (i.e., incivility) had taken on the organization and its writers. Frequently noted in these descriptions was the trend toward comment removal in the wider landscape of online news websites. The statements often moved on to a discussion of other ways that readers are able to provide commentary on news articles, with an emphasis on social media platforms as alternative outlets for the continuation of constructive conversation. Another frequently appearing theme emphasized how organizations inherently valued their online community of readers and their honest feedback and criticism, which news outlets viewed as enhancing journalistic coverage.

Our analysis identified three major themes in the way news organizations framed their decisions to remove or limit commenting functions on their online platforms: Determinations of good and bad commenting practices, shifting user engagement to social media, and news organizations' role in moderating truth (and misinformation) online. Each of these themes is discussed in further detail below.

### 3.1. Killing the Comments

Twelve of the fifteen statements gave a concrete reason as to why comments were being removed. These rationales ranged from the idea that bad commenters were skewing public perception of legitimate news stories, scientific fact, or the news organization itself; lack of civility among commenters; and the protection of sensitive subjects or content. Many statements incorporated a combination of these key rationales with varying emphasis. Significant weight was given to the outsourcing of commenting functions to social media; the majority of statements focused specifically on Facebook as a platform where commenting and discussion already did and ought to take place. Twitter came in as a close second, with seven references to the social media site. What was sometimes explicitly stated, but more often alluded to, was the sheer power of mass moderation and strict community guidelines imposed upon users by most social media platforms. In outsourcing the location of user commentary, news organizations not only put a buffer between themselves and potentially vitriolic conversation, they also free up resources that would otherwise be absorbed by the maintenance of their own comment sections.

> "*At Above the Law, given our small staff, the intensive resources required for fair and effective moderation, and the human toll moderation takes on the moderators, we decided it wasn't worth the trouble. We'd rather devote our time and energy to working on our stories and interacting with readers on social media—which has the added benefit of evangelizing for our site, increasing our Facebook likes and Twitter followers, and driving traffic to ATL through Facebook and Twitter referrals*". (Above the Law)

> "*It is no longer a core service of news sites to provide forums for these conversations. Instead, we provide the ideas, the fodder, the jumping off point, and readers take it to Facebook or Twitter or Reddit or any number of other places to continue the conversation*". (The Week)

Another revealing element of the statements was the various ways that writers and journalists chose to name the removal of commenting functions. Many simply named it as such—comment removal—while others embellished the action, saying that the publication was going to kill, scrap, scale back, discontinue, shut off, switch off, turn off, end, close, or lose their comment section. Euphemistic language about comment function removal had a

high variability among the statements and policies analyzed, with some writers choosing language that dramatized or minimized the decision.

In addition to the fifteen published statements, five organizations referenced the removal or limiting of commenting functions in their commenting policies. These policy statements were included in the dataset to offer a different perspective on the rules and regulations news organizations put in place regarding commenting; they also shared a topic in common with several of the fifteen published statements—allowing comments on some types of articles and not others. For the most part, many news websites like The Guardian, The Washington Post, and The New York Times banned comments on strictly news-related articles, while opinion and popular culture articles remained open to platform-hosted reader discussion. Parsing out the difference between articles which are good candidates for commentary and those that are not is a notable exercise for news organizations and may reveal more about how news sites and journalists understand their role as information curators. Rather than running the risk of housing untoward statements by users or spreading misinformation, these prudent news organizations have made the executive decision that user commentary, though valuable, is better left to more frivolous, less journalistically intensive topics relegated to the culture and opinion sections.

> "*At CNN,* comments *on most stories were disabled in August. They are selectively activated on stories that editors feel have the potential for high-quality debate—and when writers and editors can actively participate in and moderate those conversations*". (CNN)

### 3.2. Judging Value: Good and Bad Comments

As mentioned above, all of the statements and policies analyzed included some language which indicated good commenting practices, bad commenting practices, or some implicit diagnosis of either category. These value judgments indicate discursive strategies that, on the one hand, seek to acknowledge constructive commenters, while at the same time delegitimizing user commentary, in general. The latter strategy, in turn, helps to legitimize news organizations' decisions to abandon or outsource user comments.

Though the quantity of different words and phrases used to describe bad commenting practices outweighed descriptions of good commenting practices, the latter was still written about with some emphasis. Perhaps most interesting was the prevalence of the assertion that good comments ought to be intelligent. This word was frequently accompanied by other adjectives including lively, insightful, or constructive, all of which amount to an appeal to the commenting community to pursue some kind of high-mindedness in what they write. News organizations thrive off of many and different people viewing them as a source of reliable information, as a way to learn more about the world, yet where comments are concerned, the audience is expected to have a well-rounded, even-tempered, thoughtful debate prepared. News organizations' language about good commenting amounts to a rough sketch drawn by these delineations which take on the form of the organization's ideal reader, regardless of the organization's actual readership. However, many statements did hold space for the necessity and desirableness of thoughtful critique as components of good commenting practices.

> " ... *we are* choosing *now to elevate respectful, intelligent discourse and argument. We want smart and critical readers to have a more visible role on our site, and we're looking forward to hearing from you, and publishing you*". (The Atlantic)

> "*Of* course, *at the Daily Dot, we would like to see a more civil, compassionate Web, but we want to be careful that in the name of fostering civility, we do not inadvertently kill all dissention. It is the cacophony of the Web—the voices from every point in the spectrum that give it its vibrancy—that make it the community we love*". (The Daily Dot)

Despite some recognition of constructive commentary, the statements leaned more heavily on arguments that delegitimize commenting practices, and thus justify the decisions to abandon or outsource comments. The most common themes in language regarding poor commenting practices revolved around prejudicial statements, with specific references to

misogyny, racism, and more general abuse being widely referenced. Types of comments and commenters which were largely deemed as undesirable were often named as either trolls or spam/bots. Though five individual statements referenced the scourge of trolls in their comment sections, none defined the term in any explicit way.

> "*We've also made the not-unrelated decision to close our comments section. Over the years, robust conversation in The Atlantic comments section has too often been hijacked by people who traffic in snark and ad hominem attacks and even racism, misogyny, homophobia, and anti-Muslim and anti-Jewish invective*". (The Atlantic)

Another key bad commenting practice discussed in the statements was that which equates to what the Guardian's Stephen Pritchard termed "author abuse" (Pritchard 2016), a rationale that aligns with Zhou et al.'s (2008) definition of civil discourse as avoiding personal attacks against other users and journalists. Some statements framed the reasoning for removing comment sections as a way to take a stand against readers who share prejudicial sentiments specifically directed at the journalists responsible. Though discussions about protecting sensitive subjects and other commenters were also included in some statements, the scales were tipped slightly toward the protection of journalists.

> "*We welcome strong opinions and criticism of our work, and do not hesitate to approve critical comments. However, personal attacks against our staff will not be permitted, and any criticism should relate to the article in question*". (New York Times)

A clear example of the tension between encouraging productive commentary and community, while at the same time dismantling or fully disabling the system that allows them, is evident in the VICE statement on comment removal: "we truly value thoughtful comments and critiques from readers", Smith (2016) wrote, "and our biggest worry in killing [the comments] section was that the people who have constructive and intelligent things to say would consider this a slight against them". Smith's (2016) description of the types of comments and commenters that are legitimate, and thus to be encouraged, is a very specific and, indeed, subjective one. It is not difficult to imagine a situation in which a left-leaning news outlet finds a conservative viewpoint in a comment less than "intelligent", and vice versa.

## 4. Discussion

This analysis made legible the value judgements and expected norms news organizations and journalists assign to online user commentary. By reviewing comment removal statements in the aggregate, the patterns of language used and themes addressed in these statements revealed that these news organizations felt that they were up against the same enemy, eventually finding strength in numbers in a maneuver to best unsavory or harmful comment sections.

Unlike the more utopic view that civil commentary will in some way contribute to the health of the public sphere and deliberative democracy, Quandt (2018) coined the term "dark participation" to capture the social reality that a great deal of user commentary is motivated by or characterized as intentional misinformation, hate campaigns, trolling, and cyberbullying—all of which contribute to worries about widespread social and political polarization. In fact, research suggests that "angry" people are more likely to engage in social media discussions about politics (Wollebæk et al. 2019). Quandt argues that these "dark" forms of public engagement with the news are equally deserving of our scholarly attention, even if they seem to be used as justification for decisions by news organizations to abandon their commentary features.

In addition to these "bad" commenting practices, our analysis also sheds light on organizational perceptions of what constitutes "good" commentary. These characterizations described above align with existing conceptual understandings of civil discourse. For instance, Herbst (2010) offers a useful conceptualization of civility as "constructive engagement with others through argument, deliberation, and discourse" (p. 19). Yet, it

seems that for these organizations their concerns about poor quality comments weighed more heavily in their decisions to abandon/limit commentary functions.

By removing commenting functions from their own platforms, news organizations were able to set the terms of engagement and barriers to access. They were also able to make public their normative expectations of their ideal readership, praising high quality commenting and admonishing the poor, whether or not those attributes were taken to heart or taken to social media. Repeated references to "killing" the comments echo the notion of killing a story in newsrooms, a classic illustration of the presumption of authority over news and information. Taken together, it seems that news organizations that favor abandoning commentary features, rather than actively engaging or policing commenters, still embrace their role as gatekeepers, even if functionally they are giving up control over comment spaces. To the latter point, the decision to "kill" the comments, or relegate them to less serious news (e.g., popular culture articles) and/or third-party social media platforms, is an enactment of authority and control over the relative ability of users to engage in the journalistic process through commenting. Collectively, these efforts seem aimed at decoupling comments from core journalistic work (Ksiazek and Springer 2020). In doing so, these organizations seem to be distancing themselves from the work and resources involved in hosting and moderating user commentary, as well as the potential negative impact of poor commenting practices.

A key theme that emerged from this analysis was news organizations' outsourcing of commentary to social media. Using their extant social media presences as a scaffold for the migration of online commentary functions from in-house moderation and presentation to various other platforms not only makes would-be commenters beholden to the user agreements of the likes of Facebook and Twitter, it also allows the news organizations to transfer liability and management of whatever hostility or incivility may take place within the comments regarding any news story they publish. The choice to emphasize social media as the ideal venue for user commentary outsources the responsibility for the content of user comments and the regulation of commentary standards, as well as the moderation of user comments and the resources necessary to do so. Moreover, by encouraging users to continue to comment, albeit on platforms like Facebook and Twitter, these organizations might stand to gain from increased referral traffic generated by these social platforms.

Interestingly, the news outlets by and large did not disclose reflections or hesitations about redirecting their readers to commercially oriented social media companies. Tapping into the narrative of uncivil user commentary—which constitutes a threat to the well-being of authors, (the brands of) news outlets, and healthy public discussions—helps to legitimize the current trend of news outlets giving up the maintenance of their infrastructural role of hosting user discussions. And even more relevant, that they willingly assign this role to especially one proprietary platform which "filters our daily communicative acts through a profit-extracting sieve, deploying its intimate view of users' activities and relationships for the benefit of advertisers and others, who in turn provide further data (via the API) for the Facebook social graph" (Plantin et al. 2018). The news outlets thereby contribute to Facebook's growing "power to shape our communication behavior for its own ends . . . ". As Plantin et al. (2018) note, a handful of tech giants harvest "the power of platforms—which hold undeniable benefits for both users and smaller, independent application developers—to gain footholds as the modern-day equivalents of the railroad, telephone, and electric utility monopolies of the late 19th and the 20th centuries" (pp. 306–307). While other third-party commenting platforms exist, such as Coral, their user experience is not yet identical to Facebook's (e.g., Stroud et al. 2020).

Although novel in offering insights into organizational rationales for abandoning or outsourcing commentary functions, this study is not without limitations. The small, purposive sample is limited in generalizability and the organizations included in the analysis are primarily US-based, with an additional few based in the UK. Expanding this research globally, across varied media systems and cultures, would offer a more comprehensive understanding of the reasoning behind this recent trend in digital journalism. Additionally,

the corpus of statements used in the analysis was limited to publicly available statements from these news organizations. Semi-structured interviews with members of these, and other, organizations could add depth to our understanding of their rationales. For instance, future research could explore policies/practices concerning the inclusion or highlighting of "good" commentary, beyond the implicit diagnosis identified in this study. Additionally, interviews and focus groups with highly engaged commenters, both those who post "civil" and "uncivil" content could be conducted in order to better understand the motivation and rationale for their mode of discussion. Finally, future research could explore the legal risks and implications of managing, removing, and/or outsourcing commentary features.

Rather than abandon commenting functions, we would urge news organizations and researchers to explore more productive ways to encourage constructive commenting. The statements analyzed in this study often referenced how the organizations valued their communities of users, despite their actions (i.e., abandoning/limiting commentary) that might suggest otherwise. The tension between a desire to encourage an active community of users and devoting/managing the resources necessary to do so in a productive way was pervasive in the organizational statements. Comment management and moderation are resource-intensive for news organizations (Goodman 2013), and are often viewed as outside the purview of traditional journalistic practices.

Perhaps, rather than abandoning user commentary, news organizations could explore more efficient ways to manage these spaces. As we discussed earlier, efforts are underway to develop artificial intelligence and machine learning approaches to identify and feature productive user contributions. Related, Ksiazek and Springer (2020) advocate for an "Integrated Comment Moderation Model" that captures best practices and collaboration among user-driven moderation, organizational policies for encouraging more productive commentary, and innovations in the areas of automation in moderation.

User comments offer citizens an opportunity to actively participate in public discussion of current events, knowledge creation, and the journalistic process. Abandoning or limiting these capabilities compromises that opportunity for user engagement. By seeking ways to encourage, rather than limit, user commentary, news organizations can continue to embrace a more participatory and inclusive form of digital journalism.

**Supplementary Materials:** The following are available online at https://www.mdpi.com/article/10.3390/journalmedia2040034/s1, The codebook used (Table S1) to sort language and emergent themes within the dataset of comment removal statements and policies is included as a supplement.

**Author Contributions:** Conceptualization, T.B.K. and N.S.; Methodology, M.N.N.; Formal Analysis, M.N.N.; Resources, T.B.K. and N.S.; Data Curation, T.B.K. and M.N.N.; Writing—Original Draft Preparation, M.N.N.; Writing—Review and Editing, T.B.K., N.S. and M.N.N.; Visualization, T.B.K., N.S. and M.N.N.; Supervision, T.B.K.; Project Administration, T.B.K.; Funding Acquisition, T.B.K. All authors have read and agreed to the published version of the manuscript.

**Funding:** This research was funded by the Villanova University Summer Research Grant.

**Institutional Review Board Statement:** Not applicable.

**Informed Consent Statement:** Not applicable.

**Data Availability Statement:** Comment removal statements and policies are publicly available at the web sites listed in our references.

**Conflicts of Interest:** The authors declare no conflict of interest.

## Note

[1] The use of "stemming" here allows for search terms to produce results that include all forms of words beginning with the stems (i.e., "remov" and "comment"). For instance, both "remove" and "removal" would be included, as would "comment", "comments", "commenting", etc.

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
