# Peer review of "Killing the Comments: Why Do News Organizations Remove User Commentary Functions?"

_journalmedia, doi:10.3390/journalmedia2040034_

Round 1
Reviewer 1 Report
The article presented is of appropriate quality to be published in your magazine. The subject matter is in accordance with the theme of the magazine and it is also original and clearly current. The relationship of the media with the users but also with the truth is a very important current challenge.
From the point of view of the methodology, the selection of the sample is well described and seems adequate, although it would be appropriate to have some indication of the percentage of media that have stopped publishing comments to get an idea of the relevance or trend, as well as motivation, which is adequately described.
From our point of view the article could be published, but it should consider two aspects that we consider relevant:
1. in the first place, the relation of the comments with the possible social and political polarization. It would be appropriate that both in the introduction and in the discussion it was highlighted because it is part of the context in which social networks and the media have to deal with the problems of content management.
2. Although fake news appears as a keyword, we believe that few is reported in the article about the importance of false content that comments can contribute. In addition, it would be interesting to consider whether it is also a legal risk to avoid responsibility for the content due to the difficulty of properly managing the comments, neither in the form of curation or via algorithm.
Finally, a reflection on whether or not the media has the same problems as social networks could be interesting at the platform level. That is, in the case of social networks, the content of the users is the center of activity, but this is not the case in the case of the media. Deleting comments is an impossibility of being able to manage the contents properly and responsibly. What can be the cost for the media? It would be interesting to see statements from media that have decided to keep comments in order to have a complete picture of the media ecosystem
Author Response
We are deeply grateful to the reviewers for providing thoughtful and constructive feedback on this manuscript and appreciate the decision by Editor Ilic to allow us the opportunity to address these comments in a revision.
Both reviewers rightly suggest a number of revisions that will strengthen the manuscript. Below we respond to each comment individually.
Reviewer 1
Comment
From the point of view of the methodology, the selection of the sample is well described and seems adequate, although it would be appropriate to have some indication of the percentage of media that have stopped publishing comments to get an idea of the relevance or trend, as well as motivation, which is adequately described.
Response
Comment
From our point of view the article could be published, but it should consider two aspects that we consider relevant:
1. in the first place, the relation of the comments with the possible social and political polarization. It would be appropriate that both in the introduction and in the discussion it was highlighted because it is part of the context in which social networks and the media have to deal with the problems of content management.
Response
We added the following paragraph to the introduction (page 2, line 46):
All of this suggests a desire among news professionals to avoid further contributing to widespread social and political polarization across the globe. While some research suggests that commenters are “inclined to seek politically dissimilar conversational partners” (e.g., Liang, 2014, p. 487),” a pattern that contributes to viewpoint diversity in comment spaces, these platforms also have the potential to enable group polarization, especially if commenters act in uncivil ways toward one another.
We revised the following text (in bold) on p. 9 (line 397):
Unlike the more utopic view that civil commentary will in some way contribute to the health of the public sphere and deliberative democracy, Quandt (2018) coined the term “dark participation” to capture the social reality that a great deal of user commentary is motivated by or characterized as intentional misinformation, hate campaigns, trolling, and cyberbullying—all of which contribute to worries about widespread social and political polarization.
Comment
2. Although fake news appears as a keyword, we believe that few is reported in the article about the importance of false content that comments can contribute.
Response
We removed “fake news” from the list of keywords and replaced it with “news comment removal.”
Comment
In addition, it would be interesting to consider whether it is also a legal risk to avoid responsibility for the content due to the difficulty of properly managing the comments, neither in the form of curation or via algorithm.
Response
The legal side of this is outside of our areas of expertise, so we included this suggestion as a future research opportunity (p. 11, line 471):
Finally, future research could explore the legal risks and implications of managing, removing, and/or outsourcing commentary features.
Comment
Finally, a reflection on whether or not the media has the same problems as social networks could be interesting at the platform level. That is, in the case of social networks, the content of the users is the center of activity, but this is not the case in the case of the media. Deleting comments is an impossibility of being able to manage the contents properly and responsibly. What can be the cost for the media? It would be interesting to see statements from media that have decided to keep comments in order to have a complete picture of the media ecosystem.
Response
Both reviewers included a comment about policies/practices regarding keeping comments. During the analysis, there were no explicit policies or statements describing the inclusion criteria for user comments. However, as we note in our Results section, many of the statements did include “implicit diagnosis” of good commentary (not just bad). This would also be an interesting topic for additional future research. We now include this suggestion (in bold) in the paragraph on p. 11 (line 467):
…Semi-structured interviews with members of these, and other, organizations could add depth to our understanding of their rationales. For instance, future research could explore policies/practices concerning the inclusion or highlighting of “good” commentary, beyond the implicit diagnosis identified in this study.
Reviewer 2 Report
This is a timely and well written article on editorial practices and impact of content removal from online presence of newspapers. There are a couple of references that may need updating such as :
Liu, J., & McLeod, D. M. (2021). Pathways to news commenting and the removal of the comment system on news websites. Journalism, 22(4), 867-881.
There may also be need for a short paragraph on fact checking and decining what not to remove as an editorial practice
Author Response
We are deeply grateful to the reviewers for providing thoughtful and constructive feedback on this manuscript and appreciate the decision by Editor Ilic to allow us the opportunity to address these comments in a revision.
Both reviewers rightly suggest a number of revisions that will strengthen the manuscript. Below we respond to each comment individually.
Reviewer 2
Comment
This is a timely and well written article on editorial practices and impact of content removal from online presence of newspapers. There are a couple of references that may need updating such as :
Liu, J., & McLeod, D. M. (2021). Pathways to news commenting and the removal of the comment system on news websites. Journalism, 22(4), 867-881.
Response
We updated the Liu & McLeod (2021) reference. We also updated the following references:
Liang, H. (2014). The organizational principles of online political discussion: A relational event stream model for analysis of web forum deliberation. Human Communication Research, 40(4), 483–507. doi: 10.1111/hcre.12034
Post, S., & Kepplinger, H. M. (2019). Coping with audience hostility. How journalists’ experiences of audience hostility influence their editorial decisions. Journalism Studies, 20(16), 2422-2442.
Ziegele, M., & Jost, P. B. (2020). Not funny? The effects of factual versus sarcastic journalistic responses to uncivil user comments. Communication Research, 47(6), 891-920.
Comment
There may also be need for a short paragraph on fact checking and deciding what not to remove as an editorial practice.
Response
We’re unsure of the context for the fact checking paragraph. This wasn’t something that came up in our analysis.
As for the second point, both reviewers included a comment about policies/practices regarding keeping comments. During the analysis, there were no explicit policies or statements describing the inclusion criteria for user comments. However, as we note in our Results section, many of the statements did include “implicit diagnosis” of good commentary (not just bad). This would also be an interesting topic for additional future research. We now include this suggestion (in bold) in the paragraph on p. 11 (line 461):
…Semi-structured interviews with members of these, and other, organizations could add depth to our understanding of their rationales. For instance, future research could explore policies/practices concerning the inclusion or highlighting of “good” commentary, beyond the implicit diagnosis identified in this study.